

# Stratospheric Air Sub-sampler (SAS) and its application to analysis of $\Delta^{17}O(CO_2)$ from small air samples collected with an AirCore

Dorota Janina Mrozek[1], Carina van der Veen[1], Magdalena E. G. Hofmann[1], Huilin Chen[2,3], Rigel Kivi[4], Pauli Heikkinen[4], and Thomas Röckmann[1]

[1]Institute for Marine and Atmospheric research Utrecht (IMAU), Utrecht University, Princetonplein 5, 3584CC Utrecht, The Netherlands
[2]Centre for Isotope Research (CIO), Energy and Sustainability Research Institute Groningen, University of Groningen, The Netherland
[3]Cooperative Institute for Research in Environmental Sciences (CIRES), University of Colorado, Boulder, Colorado, USA
[4]Finnish Meteorological Institute (FMI), Arctic Research, Sodankyä, Finland

*Correspondence to:* D.J. Mrozek (dorota.mrozek@gmail.com)

**Abstract.**

We present the setup and a scientific application of the Stratospheric Air Sub-sampler (SAS), a device to collect and to store the vertical profile of air collected with an AirCore (Karion et al., 2010) in numerous sub-samples for later analysis in the laboratory. The SAS described here is a 20 m long 1/4 inch stainless steel tubing that is separated by eleven valves to

divide the tubing into ten identical segments, but it can be easily adapted to collect more or less smaller or larger samples. In the collection phase the SAS is directly connected to the outlet of an optical analyzer that measures the mole fractions of $CO_2$, $CH_4$ and CO from an AirCore sampler. The stratospheric part (or if desired any part of the AirCore air) is then directed through the SAS. When the SAS is filled with the selected air, the valves are closed and the vertical profile is maintained in the different segments of the SAS. The segments can later be analyzed to retrieve vertical profiles of other trace gas signatures that

require slower instrumentation. As an application, we describe the coupling of the SAS to an analytical system to determine the $^{17}O$ excess of $CO_2$, which is a tracer for photochemical processing of stratospheric air. For this purpose the analytical system described by Mrozek et al. (2015) was adapted for analysis of air directly from SAS. The performance of the coupled system is demonstrated for a set of air samples from an AirCore flight in November 2014 near Sodankylä, Finland. The standard error for a 25 mL air sample at stratospheric $CO_2$ mole fraction is 0.56 ‰ (1 $\sigma$) for $\delta^{17}O$ and 0.03 ‰ (1 $\sigma$) for both $\delta^{18}O$ and $\delta^{13}C$.

Measured $\Delta^{17}O(CO_2)$ values show a clear correlation with $N_2O$ in agreement with already published data.

## 1 Introduction

Monitoring and studying the distribution of greenhouse gases throughout the atmospheric column is an important constituent of understanding contemporary climate change. Carbon dioxide ($CO_2$) is the most important molecule of the atmospheric carbon cycle and the increase of its abundance in the atmosphere is the primary factor of recent radiative forcing (Ciais et al., 2013).

In the stratosphere the $^{17}O$ excess of $CO_2$ (expressed as $\Delta^{17}O(CO_2)$) is a valuable long-lived tracer for stratospheric chemistry and atmospheric circulation patterns. Measurement of the mole fraction and oxygen isotopic composition of $CO_2$ provides



information on both transport times and photochemical lifetimes of $CO_2$ (Boering, 2004; Wiegel et al., 2013). Over the past decades several sampling campaigns have been carried out to observe and to understand the oxygen isotope enrichments of stratospheric $CO_2$ (Thiemens et al., 1995; Alexander et al., 2001; Lämmerzahl et al., 2002; Boering et al., 2004; Kawagucci et al., 2008; Wiegel et al., 2013). Unfortunately, measurement campaigns with rockets, balloons and aircraft platforms are

expensive and therefore it is in general difficult to obtain stratospheric samples.

The simple and lightweight sampling system AirCore (Karion et al., 2010) provides new opportunities to sampling the high altitude atmosphere at relatively low cost. The AirCore device consists of a long (usually 100 m of longer) piece of coiled stainless steel tubing that is lifted to the stratosphere on a balloon with one end open and the other end closed. During ascent, the AirCore empties because of the decrease in pressure and during descent, ambient air successively fills the AirCore coil

again as pressure increases. The atmospheric profile information in the coil is preserved because of limited gas diffusion inside the long tube. This means that the altitude profiles of various trace gases as such as $CO_2$, $CH_4$ and CO can be determined by processing the content of the AirCore through a fast analytical system quickly after recovery of the sampler.

Previously, the air from an AirCore was vented after analysis with a real time gas analyzer, and not used for other, more sophisticated and slower analyses. We developed a Stratospheric Air Sub-sampler (SAS) that collects (the stratospheric fraction

of) air from an AirCore directly after online analysis and stores it in different segments of the SAS so that the profile is preserved. The SAS can be easily transported and processed, for example for the isotopic composition of trace gases, or for halocarbon analysis. The limitation of the method is that the analytical system must be capable of analyzing very small air samples, since the total stratospheric fraction of the AirCore profile is only of the order of 250 mL at ambient temperature and pressure, which is split into multiple segments in the SAS.

For the work presented here we apply the SAS concept to measurement of the $^{17}O$ excess of $CO_2$ in the stratospheric sub-samples. The analytical system described in Mrozek et al. (2015) was modified to allow air from the SAS to be flushed directly by the reference air into a new sample introduction unit at ambient pressures. The change of the oxidation reagent from $CeO_2$ powder to CuO wires reduced peak broadening and allowed detection of the equilibrated $CO_2$ peak without focusing on liquid nitrogen trap. The improved system is fully automated, only the change of the individual SAS segments is made manually.

The successful coupling of the SAS and the $^{17}O$ analysis system was demonstrated by $CO_2$ stable isotope measurements on stratospheric air obtained from an AirCore flight near Sodankylä, Finland in November 2014.

We use the common delta notation to quantify isotopic composition, $^i\delta = {^iR_{SA}}/{^iR_{ST}} -1$, where $^iR$ represents the heavy-to-light isotope ratio $^{17}R = (^{17}O/^{16}O)$ or $^{18}R = (^{18}O/^{16}O)$ of a sample (index SA) or international standard (index ST). $\delta$ values are expressed in ‰. The international reference material for oxygen isotopes is Vienna Standard Mean Ocean Water

(VSMOW) with $^{17}R_{VSMOW} = 382.7\ ^{+1.7}_{-2.1}) \times 10^{-6}$ (Kaiser, 2009) and $^{18}R_{VSMOW} = 2005.2 \pm 0.45 \times 10^{-6}$ (Baertschi, 1976). For quantifying the $^{17}O$ excess of $CO_2$ we use the exponential definition $\Delta^{17}O = [1 + \delta^{17}O]/[1 + \delta^{18}O]^\lambda - 1$ with $\lambda$ of 0.528, but note that also other definitions are in use (Assonov and Brenninkmeijer, 2005; Kaiser, 2009).



## 2  Method

The stratospheric air samples are collected in a two-step procedure. First, air samples from the surface up to the ceiling altitude of a stratospheric balloon flight (typically 30 km) are obtained with an AirCore system. Second, the stratospheric part of the collected air samples is recovered into a Stratospheric Air Sub-sampler (SAS) after online analysis of trace gas concentrations.

The details of the first step (Chen et al., In preparation.) are here briefly described in Sect 2.1. Sect 2.2 and Sect 2.3 provide the information relevant to the stratospheric air samples: the sub-sampling method and the description of the analytical system used for $CO_2$ isotope measurement.

### 2.1  Stratospheric air sampling with AirCore

The AirCore device that is operated by the University of Groningen (UG) and launched by the Finnish Meteorological Institute

(FMI) is made of stainless steel tubing coated with SilcoNert®1000 and has a total length of 100 m (40 m 1/4 inch and 60 m 1/8 inch, wall thickness 0.01 inch). The payload that includes the AirCore and a radiosonde (Vaisala, type RS92-SGPL) weighs about 3.6 kg. Before each flight the AirCore is filled with a standard dry fill-gas with known $CO_2$, $CH_4$ and CO mole fractions ($CO_2$ = 386.10 ± 0.09 ppm; $CH_4$ = 1880 ± 2 ppb; CO = 7972 ± 5 ppb) and closed with a shut-off valve at the inlet (Swagelok, part number SS-1GS4). A stratospheric balloon (Totex, type Tx3000) is used to launch the payload high into the stratosphere

near Sodankylä, Finland. Just before launching the AirCore, the shut-off valve is opened, so that during the ascent the fill-gas leaves the AirCore coil due to the drop in pressure. After reaching an altitude of approximately 30 km, the balloon bursts, and the descent of the payload on a parachute begins. Ambient air flows into the AirCore and the air from higher altitudes is continuously compressed and pushed towards the closed end of the AirCore by air from lower altitudes. The shut-off valve is closed automatically about 10 seconds after the landing, and the AirCore is quickly recovered and transported to the FMI

laboratory for analysis.

### 2.2  Sub-sampling into SAS

The vertical profiles of $CO_2$, $CH_4$ and CO mole fractions are measured within 2-3 hours after landing of the AirCore at FMI using a gas analyzer (Picarro, model G2401), as presented in Fig. 1. The originally closed end of the AirCore coil is connected to the analyzer, while the end with the shut-off valve is connected to the fill-gas cylinder. By opening both ends of the AirCore,

the air in the AirCore is flushed into the analyzer by the fill-gas. The flow rate is controlled at the outlet of Picarro instrument and set to 38.2 mL min$^{-1}$. Calibration air can be analyzed before and after measurement of the AirCore air. After the trace gas analysis, the top (i.e. stratospheric) part of the air collected with the AirCore (less than 20 % of the total collected air) is transferred into a Stratospheric Air Sub-sampler (SAS) that is sent to Utrecht University for $CO_2$ isotope analysis.

The timing of the sub-sampling process is based on the flow rate of the air through the analyzer and the pump. The timing

of the sub-sampling procedure had to be precisely established to ensure that the desired fraction of gas is collected with the SAS. To establish the timing for the sub-sampling procedure, we injected a spike of highly $CO_2$-enriched gas into the inlet of the Picarro analyzer and then connected the outlet of the pump to the inlet of the Picarro analyzer. Thus a closed loop without





the SAS was established, and the $CO_2$ spike was measured multiple times as it circulated through the analyzer and the pump. This experiment allowed us to determine the timing of air traveling from the inlet of the analyzer to the inlet of the SAS. The flow rate was measured with a flow meter, and the timing for the sub-sampling procedure was established. Also, the membrane pump (Picarro Inc.) that is used for the sub-sampling was carefully tested for leaks to avoid contamination of stratospheric

air with ambient laboratory air during the sub-sampling process. The outlet of the pump was connected to the gas analyzer (Picarro Inc., model G2401-m) to form a closed loop, which was filled with air with high mole fractions of $CO_2$, CO, $CH_4$. The flow rate was set to 35 mL min$^{-1}$ and the air enclosed in this closed system was circulated 9 times while measuring the $CO_2$, CO, $CH_4$ and $H_2O$ mole fractions. The $CO_2$ mole fractions stayed between 940.4 and 941.0 ppm, and a small initial fluctuation quenched after 9 cycles. The rate of change of $CO_2$ mole fraction based on these measurements with a very high

$CO_2$ mole fraction difference between the sample and ambient air was 0.1 ppm of $CO_2$ per minute. The effect of such a small contamination on the isotopic composition of $CO_2$ is therefore negligible (<0.001 ‰ given the maximum $\Delta^{17}O(CO_2)$ of 5 ‰, see below). We note, however, that CO gets contaminated in the sub-sampling process at a rate of 17 ppb min$^{-1}$ for CO (from a starting value of 500 ppb), so CO measurements would be compromised.

The SAS used for the $CO_2$ isotope measurements in Utrecht is made of 20 m long stainless steel tubing with a diameter

of 1/4 inch and rolled longitudinally to form rings. Eleven Swagelok valves (part number SS-3CXS4) named here three-way valves, divide the tubing into ten individual segments. The three-way valves are open and connect the segments when the stratospheric part of the AirCore air fills the SAS, and are kept closed when the sub-sampling process is finished. Each SAS segment contains about 25 mL of the AirCore air. Segment 1 corresponds to the highest altitude of the AirCore flight, and segment 10 contains air from the lower-stratosphere.

The idea of the Stratospheric Air Sub-sampler that is described here in detail, has already been successfully implemented to enable the radiocarbon analysis of stratospheric $CO_2$ at the Centre for Isotope Research in Groningen, the Netherlands (Paul et al., 2016).

### 2.3 Continuous flow system to measure the isotopic composition of $CO_2$

The analytical system to measure the isotopic composition ($\delta^{13}C$, $\delta^{18}O$ and $\Delta^{17}O$) of $CO_2$ in each SAS segment is based on

the principle presented in Mrozek et al. (2015). The isotopic composition is measured after gas chromatographic separation of the $CO_2$ from other air constituents. The $^{17}O$ content cannot be determined directly from the measurement at m/z = 45, because of the isobaric interferences of $^{13}C$ and $^{17}O$. Therefore, the $^{17}O$ content is obtained from isotope measurements on $CO_2$ before (Pre$CO_2$) and after oxygen isotope exchange (Post$CO_2$) with a large reservoir of oxygen. Instead of using cerium (IV) oxide ($CeO_2$) as exchange material (Assonov and Brenninkmeijer, 2001; Mrozek et al., 2015), we use in our new system

copper oxide (CuO) (Kawagucci et al., 2005). A single measurement requires two independent injections of 1 mL of air: one for direct measurement (Pre$CO_2$) and one for measurement after oxygen isotope exchange (Post$CO_2$), plus 0.7 mL for flushing the injection lines (all at 1 bar pressure). To reduce the measurement uncertainty statistically we perform multiple measurements on one air sample.




In addition to the requirement of overpressure in the injection unit, another disadvantage of the method presented in Mrozek et al. (2015) was the severe peak broadening that was introduced by the strong flow resistance of the $CeO_2$ powder in the oxygen exchange unit, which required re-focusing of the $CO_2$ after equilibration. In the new system the powdered $CeO_2$ in the oxygen exchange unit was replaced with CuO wires. As a result the peak broadening was dramatically reduced and the

equilibrated $CO_2$ could be analysed without re-focusing. In addition, we developed a custom made sample injection unit for samples provided by the Stratospheric Air Sub-sampler (SAS), where both sample and reference gas are injected via the SAS.

The improved CF-IRMS analytical system consists of: a sample injection unit to attach the SAS segments, a gas chromatographic column to separate the $CO_2$ from a 1 mL air aliquot, an oxygen isotope exchange unit for $CO_2$ equilibration with CuO, and open split interface and an IRMS for isotope measurement (Fig. 2). Similar to the method described by Mrozek et

al. (2015), three 6-port 2-position Valco valves (VICI, C6UWM) direct the gas flows through the analytical system. The four main components are described in the following subsections.

### 2.3.1 Sample injection

The ten individual segments of the Stratospheric Air Sub-sampler are measured separately (see Sect. 2.4.1). After connecting a segment, the volume between the mass flow controller and the segment is evacuated with a low vacuum pump and filled with

reference air. All connections in the injection sub-unit are made of 1/8 inch o.d., stainless steel tubing. When the valves of a connected SAS segment are opened, as demonstrated in Fig. 2, the reference air flushes the stratospheric air sample from the SAS into the analytical system. The flow rate of the mass flow controller is set to 1 mL min$^{-1}$ during sample gas injection. We use the reference air itself as the carrier for the sample air.

A single $\Delta^{17}O(CO_2)$ measurement requires two independent injections of the sample gas. The first injection is used for direct

isotope measurement of $CO_2$ (PreCO$_2$), the second injection is for measuring the isotopically equilibrated $CO_2$(PostCO$_2$). We inject the gas (reference air or sample air) into the GC column via a 1 mL sample loop in V1. The sample loop is always filled to ambient pressure because it is open to the outside air via the vent.

It takes less than 1 h to measure an individual SAS segment, however, in the present setup reference air is measured for several hours between the different segments (see Sect. 2.4.2). Usually two SAS segments are measured in one day. The

CF-IRMS system operates fully automated and the only manual step required is the connection of the segments.

### 2.3.2 $CO_2$ separation from air

This sub-unit consist of a gas chromatography (GC) capillary column (ParaPLOT Q 25 m × 0.53 mm, Varian). As a carrier we use helium at a flow rate of 4 ml min$^{-1}$, supplied to the GC through one of the ports in valve V1. The GC is kept inside a heated stainless steel box at 40 °C to ensure uniform conditions for gas separation. On the GC column the different air constituents are

separated. The air peak (mainly $O_2$ and $N_2$) elutes at 120 s after sample injection, $CO_2$ at 160 s and $N_2O$ at 190 s. The air peak leaves the analytical system through the vent in valve V3. Both $CO_2$ and $N_2O$ are directed either to IRMS, or to the isotope exchange sub-unit. Nitrous oxide in our system does not interfere with $CO_2$ during the isotope ratio measurement because it is fully separated from $CO_2$ and gets destroyed in the CuO oven. This is discussed in detail in Sect. 3.1.



### 2.3.3 Oxygen isotope exchange with CuO

This sub-unit consists of Valco valve 2 (V2) and an optional oxygen equilibration oven. The Valco valve V2 directs the $CO_2$ aliquot either directly to the IRMS or first into the equilibration oven, before entering IRMS. The oven is an assembly of a quartz glass reaction tube (1.4 mm i.d., 3.0 mm o.d., 300 mm length), a tube furnace, and a temperature controller. Inside the

tube there are oxygenated Cu wires and a Ni catalyst. We refer to this assembly as CuO oven. At a temperature of 900 °C, the wires act as a fast and highly efficient oxygen equilibration medium. This is similar to Kawagucci et al. (2005), who for the same purpose used twisted wires of CuO and Pt catalyst, and different from Assonov and Brenninkmeijer (2001) and Mrozek et al. (2015), who used $CeO_2$ powder. The important advantage of oxygenated Cu/Ni wires over $CeO_2$ powder in our system is the much smaller flow resistance. Therefore, the new exchange unit induces a much smaller peak broadening, and

the equilibrated $CO_2$ does not require focusing anymore. As a result, the single analysis time shortened by 250 s in comparison to the method described in Mrozek et al. (2015) (650 s instead of 900 s) and no liquid nitrogen is required.

Before first use, and then on a weekly basis, the CuO oven is conditioned with $O_2$ at a flow rate of 20 mL min$^{-1}$ and a temperature of 600 °C. Under these conditions, the copper metal forms a coating of copper (II) oxide on the surface of the Cu wires according to:

$$Cu + \frac{1}{2}O_2 \rightarrow CuO \tag{R1}$$

### 2.3.4 IRMS interface and mass spectrometric analysis

This sub-unit of the analytical system (Figure 2 d) consists of a Valco valve 3 (V3), a Nafion$^{TM}$ dryer, a custom-made open split system (Röckmann et al., 2003) and an IRMS (Thermo Fisher Scientific Delta V Advantage). The valve V3 allows the insertion of an additional volume of 1 mL volume into the flow path, which smoothens the $PostCO_2$ peak and improves precision of

the isotope measurement. The Nafion dryer removes traces of water before the gas stream enters the IRMS via the open split system, which is also used to inject the pure $CO_2$ working gas. The IRMS measures ion current ratios 45/44 and 46/44 that originate from $CO_2$ isotopologues with masses 44, 45 and 46.

### 2.3.5 Reference air and reference $CO_2$

Our reference air cylinder (referred to as RefAir in the following) was filled with tropospheric air collected at an altitude of 20

m from the sixth floor of the Buys Ballot building on the Utrecht University campus in July 2014. The isotopic composition of the $CO_2$ in the reference air cylinder was calibrated versus air cylinders provided by an intercomparison program of the World Meteorological Organization (WMO), and assigned the following isotope values: $\delta^{13}C_{RefAir/VPDB}$ = -8.09 $\pm$ 0.10 ‰, and $\delta^{18}O_{RefAir/VSMOW}$ = 41.05 $\pm$ 0.20 ‰. Following a mass dependent fractionation relation of tropospheric $CO_2$ (Kaiser, 2009), we assign $\delta^{17}O_{RefAir/VSMOW}$ = 21.5 ‰, so that $\Delta^{17}O_{RefAir}$ = 0.0 ‰. The isotopic composition of the reference air

is measured continuously in between the samples to monitor the stability and to correct for long-term trends of the CF-IRMS system.





The $\delta$ values for the working reference $CO_2$ that is injected via the open split system are $\delta^{13}C_{RefCO2/VPDB}$ = -36.16 $\pm$ 0.01 ‰ and $\delta^{18}O_{RefCO2/VSMOW}$ = 4.69 $\pm$ 0.01 ‰.

## 2.4 Measurement procedure

### 2.4.1 Connecting the sub-sampler to the continuous flow isotope analysis system

We start the $CO_2$ isotope analysis always from the SAS segment with the highest segment number, here segment number 10. The common port of valve no. 11 (the end of segment 10) is connected to the injection line, which leads to V1, and the free port of valve no. 10 (the beginning of segment 10) to the MFC delivering reference air (see Fig. 2). Note that the beginning of the segment 10 is also the end of segment 9.

The injection lines are open to the laboratory air when the SAS segments are being exchanged. To avoid mixing of the
precious sample air with laboratory air we evacuate the volume between the MFC and the SAS segment with the low vacuum pump (LV) and flush this volume with reference air (LV pump exchanged to vent). It is important to depressurize the volume behind the MFC before opening the SAS segment. Overpressure in the injection lines must be avoided so that the small air sample is not pushed out of the inlet system.

After connecting a SAS segment as described above, the three-way valve no. 10 is opened, so that reference air starts slowly
flowing through segment 10. Next, the three-way valve no. 11 is opened, so that air is flushed to the sample loop of the injection system (see Fig. 2). The sample admission procedure is described in detail in the next section.

As the sample air is flushed into the system by the reference air, we simply continue measuring the reference air that is then flowing through the SAS segment. A disadvantage is that the last injections of the sample air are actually a mixture of sample and reference air (see below). The measurements of reference air after the sample measurement are later used for
referencing. After the reference air measurements are completed, we close the three-way valves and disconnect segment 10. As the common port of valve no. 10 has to be connected to the injection line when measuring segment 9, segment 10 has to be physically disconnected from the SAS. Segment 9 is then treated the same as segment 10 before. We continue measuring and exchanging SAS segments one by one.

### 2.4.2 Isotope analysis procedure

Each Valco valve has two possible positions: LOAD and INJECT. The initial Valco valve configuration is shown in Fig. 2: V1 and V3 are in position LOAD and V2 is in position INJECT. After a SAS segment has been connected and the connecting lines flushed as described above, sample admission starts. The MFC is set to a flow rate of 1 mL min$^{-1}$ 15 s before opening the SAS valve in order to provide a small and reproducible overpressure at the entrance of the SAS segment. We then open the three-way valve towards the SAS segment to admit the reference air from the MFC, and 1 s later the three-way valve before V1.
Now, the flow of reference air pushes the AirCore air towards the sample loop in V1. It takes 10 s for the AirCore air to reach the sample loop. After 80 s, the flow rate at the MFC is stopped to save sample and the air is allowed to further expand into and fill the injection loop. At 90 s the loop is fully filled and Valco valve V1 is switched for 40 s to position INJECT to transfer



the first aliquot of the sample air into the analytical system. The 1 mL aliquot of air from the SAS segment is transferred to the GC column in a He carrier gas (4 mL min$^{-1}$). In the GC column, the $CO_2$ is separated from other atmospheric gases. All compounds are directed via Valco valve V2 (INJECT) and Valco valve V3 (LOAD) directly towards the isotope detection unit. The $CO_2$ is injected into the ion source of the IRMS (PreCO$_2$), all other gases are discarded via the open split.

As soon as the $CO_2$ peak (PreCO$_2$) appears on the chromatogram, the second aliquot of the AirCore air is introduced into the system. Similar to the first injection, the sample loop is flushed with the AirCore air for 80 s (MFC is set to a flow rate of 1 mL min$^{-1}$ between 290 and 370 s). At 380 s valve V1 is switched for 40 s from LOAD to INJECT, and the second aliquot of the AirCore air is injected into the GC column. Non-$CO_2$ gases leave the analytical system through the open split capillary between 500 and 520 s (V2 in position INJECT and V3 in position LOAD). At 545 s, V2 switches to position LOAD, and the
$CO_2$ is directed to the isotope exchange unit. After the isotope exchange reaction, the equilibrated $CO_2$ (PostCO$_2$) is flushed further to V3. An additional 1 mL stainless steel volume in front of Valco valve V3 has been added to smoothen the peak shape of the equilibrated $CO_2$ (PostCO$_2$), leading to improved precision. The equilibrated $CO_2$ is detected on the chromatogram between 560 and 630 s.

    Figure 3 presents an example of an IRMS chromatogram. A single measurement including two injections takes 650 s. The
$CO_2$ peak from the first injection (PreCO$_2$) is detected between 250 and 280 s, and the equilibrated $CO_2$ peak from the second injection (PostCO$_2$) between 560 and 640 s. The eight working reference $CO_2$ peaks (RefCO$_2$) are injected to the IRMS directly via the open split interface, four before detection of the PreCO$_2$ (0-200 s), and four before detection of the PreCO$_2$ (300-500 s). The two sample peaks (PreCO$_2$ and PostCO$_2$) have a different shape because the second one has passed the exchange unit and the additional volume.

For each SAS segment we run a sequence of 35 individual measurements (of two injections each). The results show that 5 of these measurements represent pure sample gas (SA), 10 of them contain sample-reference air mixtures, and 20 are pure reference air (RefAir) measurements (see Fig 4). Theoretically, we should be able to analyse 9 SA aliquots from each SAS segment (25 mL / 2.7 mL), but as described above the reference air is used as carrier gas and mixes with the sample air. Since we know the isotopic composition of the reference air, the method can potentially be improved by extracting SA information
also from the "mixed" SA/RefAir peaks, which may increase measurement uncertainty statistically (see Table 1).

## 3 Performance of the continuous flow isotope analysis system

### 3.1 Separation and destruction of $N_2O$

Nitrous oxide ($N_2O$) interferes with the mass spectrometric analysis of $CO_2$ because the $CO_2$ and $N_2O$ isotopologues fall both on the mass 44, 45, 46 collectors and cannot be separated with conventional IRMS instruments (Mook and van der Hoek,
1983). These mass interferences can lead to significant analytical biases for $\delta^{13}C(CO_2)$, $\delta^{17}O(CO_2)$ and $\delta^{18}O(CO_2)$ values, and as result decrease $\Delta^{17}O$ of stratospheric air by as much as by 3 ‰ (Wiegel et al., 2013). Note that $N_2O$ and $CO_2$ can also not be separated cryogenically (Mook and Jongsma, 1987).





To measure $CO_2$ isotopes in an air sample without interference from $N_2O$ we completely separate $N_2O$ from $CO_2$ before the isotope ratio analysis on the GC column (Ferretti et al., 2000). In addition, for the measurement of PostCO2, $N_2O$ is quantitatively destroyed in the hot isotope exchange unit (Kawagucci et al., 2005; Assonov and Brenninkmeijer, 2006; Mrozek et al., 2015).

To demonstrate successful GC separation and $N_2O$ decomposition inside the CuO oven at 900 ºC we connected a dilution of 400 ppm $N_2O$ in synthetic air to the injection sub-unit of our analytical system. The $N_2O$ enriched gas was placed at the position of a flask, as shown in Fig. 2. The flask was opened and the $N_2O$ enriched gas filled the injection lines and the 4 mL inner volume of MFC. Next, we changed the position of the three-way valve in front of the MFC so that the reference air was flowing towards the MFC. As a consequence, the reference air mixed with the $N_2O$-rich gas and both $N_2O$ and $CO_2$ were

observed on the chromatogram. We monitored the $N_2O$ peak before and after the isotope exchange reaction: for the non-heated aliquot, the GC column separated completely the $N_2O$ from the $CO_2$; for the heated aliquot, the $N_2O$ was destroyed completely in the CuO oven (see Fig. 5). For the experiment shown in Fig. 5, the chromatogram was extended to 750 s instead of the normal length of 650 s because $N_2O$ elutes after $CO_2$. Additionally, the 5th working RefCO2 peak was omitted in order to observe the $N_2O$ peak. The test shows that $N_2O$ can be effectively removed on CuO/Ni wires at 900 ºC. This removal method can

potentially be applied to other trace gas measurement techniques.

### 3.2   CuO - $CO_2$ equilibration efficiency

To quantify the efficiency of oxygen isotope equilibration in the CuO oven we prepared a dilution of RefCO2 in synthetic air in a 2 L flask and we connected the flask to the injection sub-unit of the analytical system. The RefCO2 dilution had a mole fraction similar to tropospheric air $CO_2$ (400 ppm) and $\delta^{18}O(CO_2)$ values about 36 ‰ different from the reference air (the

isotope values are given in Sect. 2.3.5). First, the reference air was injected to the system through a SAS segment and we registered 10 measurement of pure RefAir. Next, we stopped the measurement sequence to evacuate the SAS segment with the LV pump and to fill the SAS segment with the RefCO2 dilution. The measurement sequence was then continued for another 25 measurements. From the difference in the oxygen isotopic composition between RefAir and RefCO2 dilution before and after oxygen isotope exchange, the oxygen exchange efficiency in CuO oven was calculated to be >99.5 %. We conclude that the

oxygen exchange reaction with CuO/Ni wires at 900 °C is complete.

### 3.3   Error analysis

In order to investigate the long term stability of our CF-IRMS system and the measurement uncertainty, 540 aliquots of reference air were injected continuously to the CF-IRMS system for 50 hours, comprising 270 individual measurements of the complete isotopic composition of $CO_2$. The standard deviation of $\Delta^{17}O(CO_2)$ over all 270 measurements was 1.22 ‰. This is

the error that we assign to a single measurement with the new analytical system. In our previously published method (Mrozek et al., 2015), the $\Delta^{17}O(CO_2)$ standard deviation in such a long term stability test was 1.68 ‰. The improvement compared to the system described in Mrozek et al. (2015) is likely due to the replacement of the isotope exchange medium from *powdered*



CeO$_2$ to CuO *wires*, and through abandonment of a liquid nitrogen trap for re-focusing of the isotopically equilibrated CO$_2$. For $\delta^{18}$O(CO$_2$) and $\delta^{13}$C(CO$_2$) the standard deviation over all 270 measurements was 0.06 ‰ and 0.07 ‰ respectively.

To investigate how the measurement error is reduced statistically with multiple measurements on an air sample, we divided the 270 measurements from the stability test into different packages of *n* numbers of measurements (for example, 2 packages of 135 measurements each, 3 packages of 90 measurements each, etc.). The experimental standard error (SE) for each set of packages was then compared to the theoretically expected error. Calculation for $\Delta^{17}$O(CO$_2$) are shown in Table 1, the results for $\Delta^{17}$O(CO$_2$), $\delta^{18}$O(CO$_2$) and $\delta^{13}$C(CO$_2$) are displayed in Fig.7. Since the SE is defined as $\sigma/n^{0.5}$ the expected slope on this double logarithmic plot is -0.50. The error reduction follows the theoretically expected relation quite well. When the two groups with only 2 and 3 members each are neglected, the linear fit to the data has a slope of -0.46 $\pm$ 0.02 for $\Delta^{17}$O, -0.42 $\pm$ 0.03 for $\delta^{18}$O and -0.45 $\pm$ 0.03 for $\delta^{13}$C. Within the error, this is close to the theoretical slope of -0.50.

We conclude that repeated measurements on one air sample (up to 54 repetition) can reduce the uncertainty in $\Delta^{17}$O(CO$_2$) to 0.2 ‰. More than 54 repetitions on one air sample improves the $\Delta^{17}$O(CO$_2$) uncertainty only marginally. For 5 repeated measurements on one stratospheric air sample stored in the SAS (an example of a CF-IRMS measurement sequence is given in Fig. 4) we therefore expect an uncertainty of 0.57 ‰ for $\Delta^{17}$O(CO$_2$) and 0.03 ‰ for both $\delta^{18}$O(CO$_2$) and $\delta^{13}$C(CO$_2$). This will be compared to the reproducibility of the actual SAS measurements in Sect. 4.2.

## 4 AirCore flight on November 5th, 2014

The AirCore payload from the University of Groningen was launched about 100 km upstream from Sodankylä, using a meteorological balloon. The AirCore was released from the balloon at 34.5 km altitude (at 4.2 hPa) and the coil filled with air during the balloon descent. Time of launch was 10:04 UTC, while the payload landed at 12:40 UTC, thus the total flight duration was 2 hours and 36 minutes. During the balloon ascent the payload traveled south-east towards Sodankylä, due to winds in the stratosphere and troposphere. The payload landed on a parachute 53.8 km east from the analysis laboratory of the Finnish Meteorological Institute (FMI). After landing the AirCore was transported to the FMI laboratory. The analysis of trace gases started 2 hours and 15 minutes after the landing of the payload. In the laboratory the mole fractions of CO$_2$, CH$_4$ and CO were measured and the stratospheric air was transferred into the SAS. The SAS was sent to the laboratory in Utrecht for CO$_2$ isotope analysis.

### 4.1 Assigning the trace gas data to the SAS segments

We assigned the trace gas mixing ratios measured with the Picarro instrument to the individual SAS segments based on the flow rate of the carrier gas and time required to fill the SAS. Figure 7 shows which part of the stratospheric AirCore air is stored in which segment of the SAS, both as function of altitude and pressure. Note, that the SAS segments are equally spaced in pressure change (here 16 mbar/segment) during the descent rather than altitude. For example: the SAS segment 1 contains air from 24.5 to 21.4 km, while the SAS segment 2 contains air from 21.4 to 19.2 km, etc. An analysis of the trace gas profiles from the AirCore flight will be published separately.





## 4.2 $\Delta^{17}O(CO_2)$ analysis of the AirCore air stored in the SAS

The stratospheric part of the AirCore air from 5th November 2014 flight was measured in Utrecht between 25 November and 1 December 2014. The results showed that segments 10 and 8 were contaminated with $CO_2$ from outside air. The precise origin of this contamination could not be determined, but it could have happened when connecting SAS segments to the injection

sub-unit of the CF-IRMS system. Due to an accidental instability in the helium carrier gas flow, the air sample from the SAS segment 6 was lost.

As shown in Fig. 4, we are able to perform five repeated CF-IRMS measurements on the other seven stratospheric air samples stored in SAS. The average (1 $\sigma$) standard deviation of the raw molecular mass ratios (45/44 and 46/44) of an air sample is 0.04 and 0.02 ‰ for non-equilibrated $CO_2$, and 0.07 and 0.11 ‰ for equilibrated $CO_2$, respectively. The main contribution to

the measurement error of $\Delta^{17}O$ is the uncertainty in the isotope ratio 45/44 (Brenninkmeijer and Röckmann, 1998). Based on the reproducibility of m/z 45 of equilibrated and non-equilibrated $CO_2$ ( $\sqrt{(0.04)^2 + (0.07)^2} \times$ 15), the uncertainty in $\delta^{17}O$ for a single measurement is 1.25 ‰ and for a package of five measurements it is 0.56 ‰ (1.25 ‰ /$\sqrt{5}$). The error of 0.56 ‰ determined for the real AirCore air measurements agrees very well with the uncertainty of 0.57 ‰ expected from the long term stability test, see Sect. 3.3. The standard deviations for all seven successfully measured SAS segments were 0.08 ‰ for $\delta^{13}C$

and 0.05 ‰ for $\delta^{18}O$. This results in the standard error of 0.03 ‰ for $\delta^{13}C$ and 0.02 ‰ for $\delta^{18}O$ for a package of 5 repeated analyses from a SAS segment. This is again very similar to the expected error from the long term stability test as presented in Fig 6.

## 4.3 $\Delta^{17}O(CO_2)$ - $N_2O$ correlation

The mole fraction of $N_2O$ is a good tracer for the photochemical processing of long-lived trace gases in the stratosphere (Park

et al., 2004; Kaiser et al., 2006). $N_2O$ was not measured on the AirCore air, but $CH_4$ was measured and we use the relationship between stratospheric $CH_4$ and $N_2O$ mole fractions at high latitudes to translate the $CH_4$ mole fractions to $N_2O$ mole fractions. For this purpose we used $CH_4$ - $N_2O$ data from two cryosampler flights in the Arctic in 2009 and 2011 (Engel et al., 2015). The fit function for the $CH_4$ - $N_2O$ data set was provided to us by Andreas Engel, Goethe University Frankfurt, Germany. The stratospheric pseudo-$N_2O$ profile derived this way was then averaged according to the ten individual SAS segments.

In Fig. 8 we show the $N_2O$ mole fractions deduced from the $CH_4$ profile versus the $\Delta^{17}O(CO_2)$ data of the AirCore air samples analysed from the SAS. The linear negative correlation between $N_2O$ and $\Delta^{17}O(CO_2)$ is well known from previous studies and was discussed before in (Boering, 2004; Kawagucci et al., 2008; Wiegel et al., 2013). Here, we use this correlation as an independent, additional check on the system calibration. The AirCore air samples agree with the already published data, suggesting that our measurements are well calibrated. The scatter of our data is similar to the variability of the data presented

by Kawagucci et al. (2008). The data presented by Wiegel et al. (2013) show smaller variability, but our measurements were obtained from samples of 25 ml of air at ambient pressure only; this is up to 80 times smaller sample size than used in the measurements of Wiegel et al. (2013). The variability of our data around the linear fit is within the analytical uncertainty of 0.56 ‰ (1 $\sigma$) per SAS segment. The uncertainty of the $N_2O$ mole fraction in Fig. 8 reflects the standard deviation of the $N_2O$





profile stored in an individual SAS segment. It is up to 20 ppb (1 $\sigma$) for the high altitude AirCore air samples where $N_2O$ showed a strong trend and 2 ppb (1 $\sigma$) for lower altitude samples where $N_2O$ is more constant. The complete $CH_4$, CO and $CO_2$ mole fraction dataset and in-depth analysis on $CO_2$ isotope measurements from the November 5th 2014 AirCore flight over Sodankylä, Finland, will be published separately.

We conclude that air sampling with an AirCore, followed by sub-sampling of stratospheric air in 10 segments of the SAS and analysis with the analytical system described here provides a relatively cost-effective technique for obtaining vertical profiles of the complete isotopic composition of $CO_2$ in the stratosphere.

## 5   Conclusions

This article describes the concept of the Stratospheric Air Sub-sampler (SAS), a device constructed to recover and store the
stratospheric part of air obtained with an AirCore sampler. The SAS described here is an assembly of 20 m, 1/4 inch o.d., stainless steel tubing divided into ten segments. Each SAS segment contains 25 mL of air sample at $\approx$ 980 mbar. During the procedure of sub-sampling air into the SAS, the air from an AirCore is measured for $CO_2$, $CH_4$ and CO mole fractions with a Picarro analyzer. This leads to some mixing of air and partial loss of the vertical information, but once stored in the SAS, the stratospheric profile of the AirCore air after online analysis is preserved and the individual sub-samples can be supplied to
relatively slow analytical instrumentation. The SAS is easy to construct and simple to use, also in the field.

To illustrate the scientific possibilities, the SAS was coupled to a continuous flow analytical system for measurement of the $^{17}O$ excess of $CO_2$. The air from the SAS was supplied via a new sample introduction system, to allow isotope analysis on very small samples. The sample injection occurs at ambient pressure and we use the reference air as the carrier gas. The determination of $\Delta^{17}O$ of $CO_2$ is performed by measuring $CO_2$ before and after complete oxygen isotope exchange with a
large oxygen reservoir provided by CuO. The standard error for a 25 mL air sample at stratospheric $CO_2$ mole fraction is 0.56 ‰ (1 $\sigma$) for $\Delta^{17}O$ and ‰ (1 $\sigma$) for both $\delta^{18}O$ and $\delta^{13}C$. Since no focusing of isotopically exchange $CO_2$ is needed the analytical system operates liquid nitrogen free.

The concept of SAS and its coupling to the analytical system for measurement of $\Delta^{17}O$ was validated through measurements on stratospheric air samples obtained during an AirCore flight over Sodankylä, Finland in November 2014. $\Delta^{17}O$ shows the
expected negative linear correlation with $N_2O$, which provides an independent check on the system calibration. In the future, we plan routine measurements of stratospheric profiles of $\Delta^{17}O$ together with the mole fractions of $CO_2$, $CH_4$ and CO from AirCore flights on a regular basis, which could expedite the use of the isotope signatures for studying stratospheric circulation patterns and reaction mechanisms. The concept of SAS will further broaden the scientific questions that can be addressed by AirCore soundings (e.g. Paul et al. (2016)).

*Acknowledgements.* This work was funded by the Marie-Skłodowska Curie ITN INTRAMIF (Initial Training Network in Mass Independent Fractionation) as a part of the European Community's Seventh Framework Program (FP7/2007-2013), Grant Agreement 237890. Research



at the FMI has been partly supported by the EU project GAIA-CLIM and the Academy of Finland grant number 140408. We thank Andreas Engel, Goethe University Frankfurt, Germany, for sharing with us the stratospheric $N_2O$ - $CH_4$ correlation function.



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





**Figures**

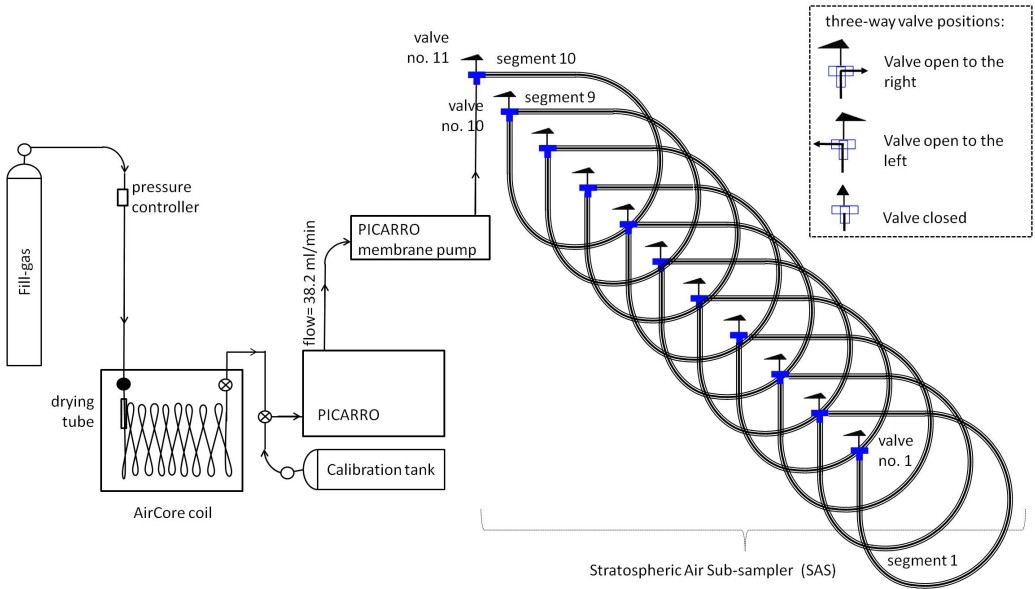

**Figure 1.** Schematic diagram of the analytical system for trace gas analyses with a Picarro instrument and for transferring the air from the AirCore coil into the Stratospheric Air Sub-Sampler (SAS). The three-way valves in the SAS are in "open to the right" position when the AirCore air is transferred into the SAS and closed after sub-sampling. The crossed circles are conventional valves. The open circles at the inlet of the fill-gas and the calibration gas cylinder represent cylinder valves and pressure regulators. The black circle represents the shut-off valve at the end of the AirCore coil. The drying tube is filled with magnesium perchlorate ($Mg(ClO_4)_2$) from Sigma-Aldrich.



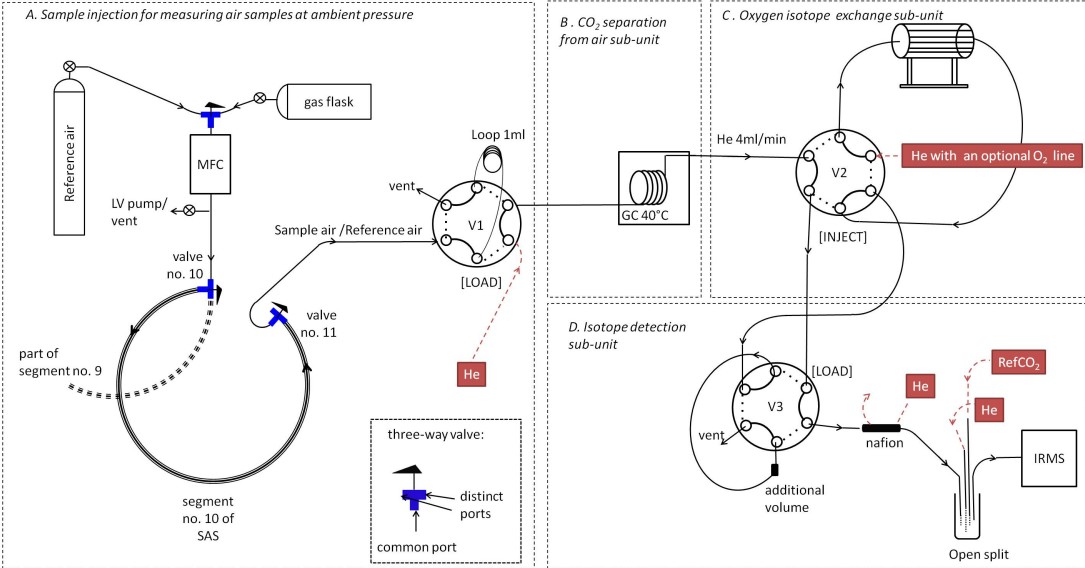

**Figure 2.** Schematic diagram of the CF-IRMS analytical system for complete isotope analysis (including $\Delta^{17}O$) of stratospheric $CO_2$. The system is divided into four units: (A) sample injection, (B) gas chromatographic column (GC) to separate $CO_2$ from air and $N_2O$ (C) optional oxygen isotope exchange unit, (D) isotope detection unit including open split interface and Isotope Ratio Mass Spectrometer (IRMS). MFC is mass flow controller, V1-V3 are Valco valves, RefCO$_2$ is Utrecht working reference $CO_2$, the crossed circles are conventional valves, and the blue T's are the three-way valves. The SAS three-way valves are normally closed but here they are shown in the sample admission position.





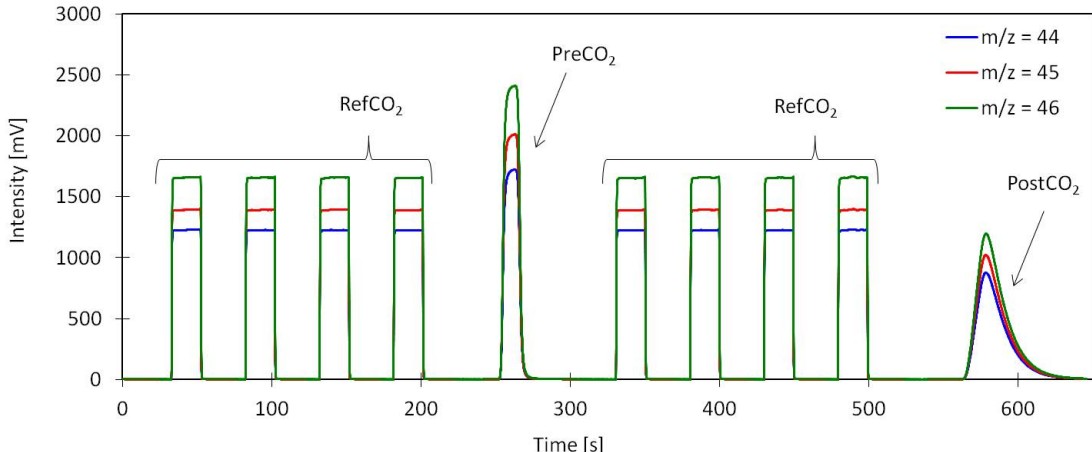

**Figure 3.** Example of a typical IRMS chromatogram of a single $\Delta^{17}O(CO_2)$ analysis with the CF-IRMS system described here. The eight square peaks are the working reference $CO_2$ (RefCO$_2$) peaks injected directly via the open split interface. The non-equilibrated $CO_2$ (PreCO$_2$) peak and the equilibrated $CO_2$ (PostCO$_2$) peak are the two sample gas peaks.





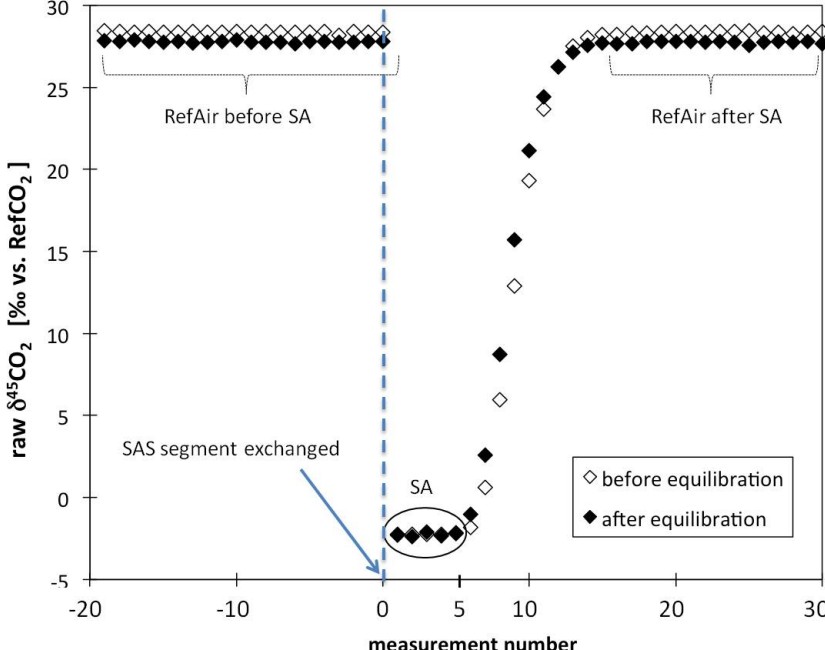

**Figure 4.** Raw-$\delta^{45}CO_2$ data [‰ vs RefCO$_2$] from a complete measurement sequence of an SAS segment. SA is the air sample stored in the SAS segment. *RefAir before SA* refers to the reference air passing through an empty SAS segment and defines IRMS stability before the new SA is connected. After exchanging the SAS segments we obtain five "pure" sample air measurements. Subsequent to the SA measurement, the reference air first mixes into the sample air and it takes about 10 measurements until pure reference air is processed. *RefAir after SA* refers to the reference air passing through the SAS segment after the SA has been completely flushed out. For the sample air presented here the "before equilibration" points overlap with the "after equilibration" points and cannot be distinguished.



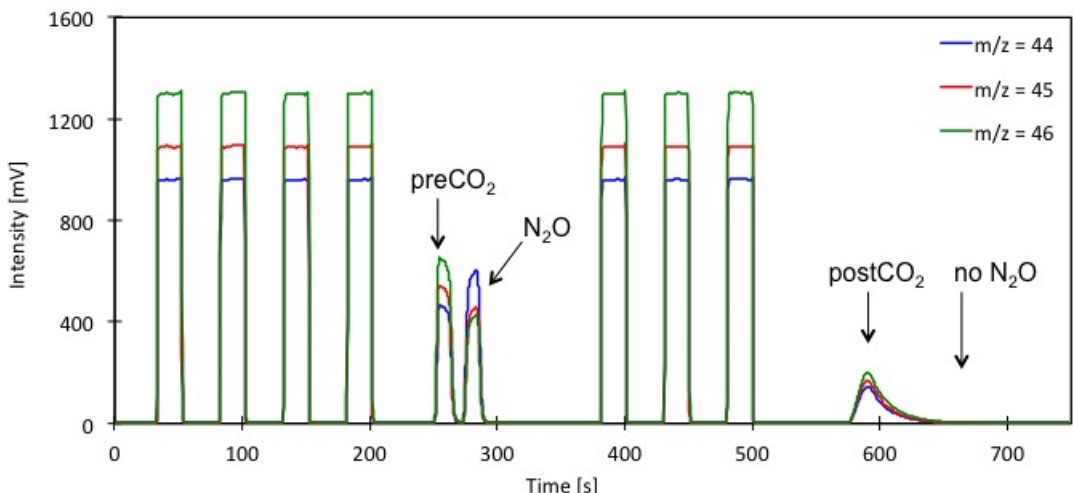

**Figure 5.** IRMS chromatogram demonstrating the separation of $N_2O$ from $CO_2$ on the GC column and the absence of an $N_2O$ peak after the hot CuO/Ni isotope exchange unit. The artificially prepared mixture of $N_2O$ and $CO_2$ was injected to the analytical system following the procedure described in Sect. 2.4.2. In order to observe the $N_2O$ peak the 5th working $RefCO_2$ peak was omitted in the chromatogram and the measurement time was extened to 750 s.





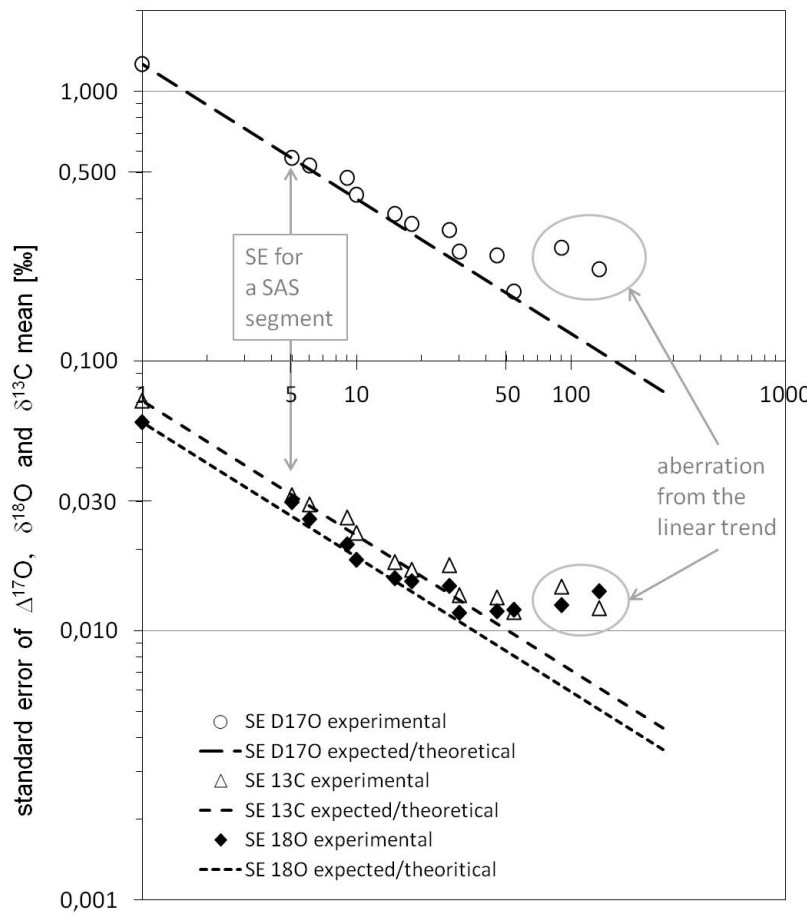

**Figure 6.** Correlation of ln(SE) versus ln($n$) for $\Delta^{17}O(CO_2)$ (based on data from Table 1), $\delta^{18}O(CO_2)$ and $\delta^{13}C(CO_2)$. SE is the standard error of the mean, and $n$ is the number of measurements in a package. The experimental SE of $\Delta^{17}O$, $\delta^{18}O$ and $\delta^{13}C$ are shown as circles, triangles and black diamonds respectively. The dashed/dotted lines are the theoretically expected slopes for the linear correlation between ln(SE) and ln($n$). For a packages of $n \le 54$, the experimentally derived slopes are -0.46 ± 0.02 for $\Delta^{17}O$, -0.42 ± 0.03 for $\delta^{18}O$ and -0.45 ± 0.03 for $\delta^{13}C$. The values are close to the theoretical slope of -0.50 (dotted / dashed lines). Based on this experiment we expect an uncertainty of ± 0.57 ‰ for $\Delta^{17}O$ of an AirCore air sample stored in an individual SAS segment.





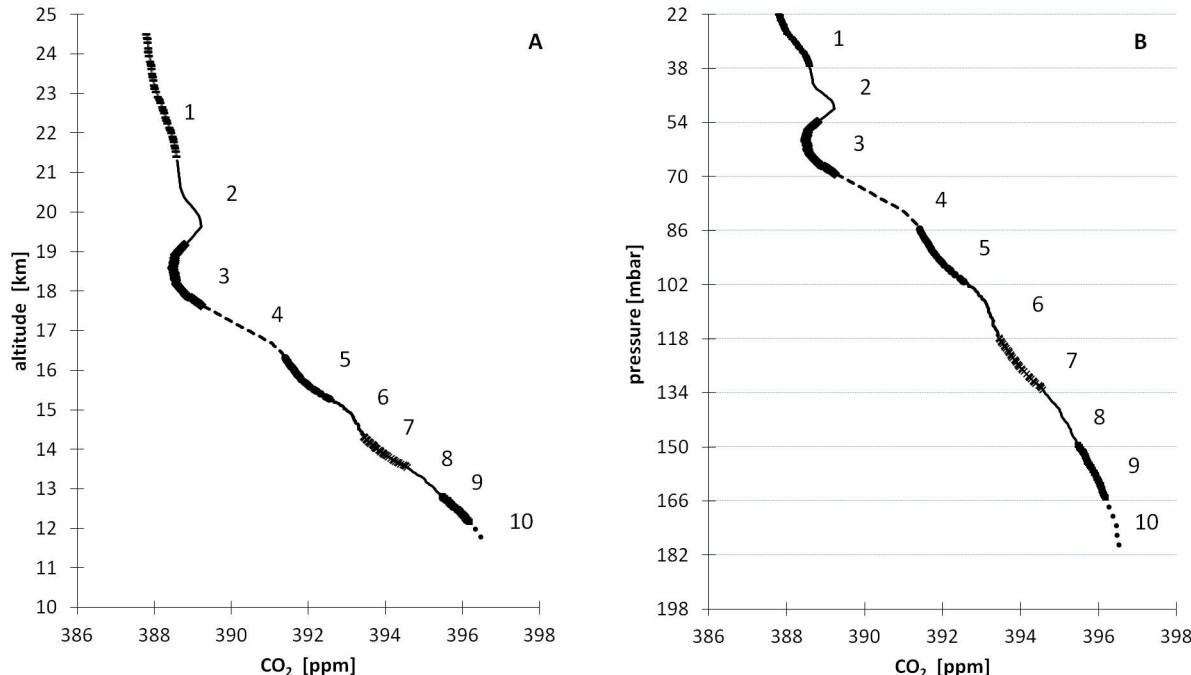

**Figure 7.** Stratospheric $CO_2$ mole fraction profile for an AirCore descent near Sodankylä, Finland (67 °N) in November 2014, as a function of altitude [km] and pressure [mbar]. The numbers and the various line styles indicate the sub-sampler segments into which the respective part of the air from the AirCore coil was transferred after the trace gas analysis. The general trend of the profile agrees with the already published data (Foucher et al., 2011). The distribution of the air sample in the SAS segments is proportional to the pressure change during the AirCore descent, here 16 mbar per SAS segment.





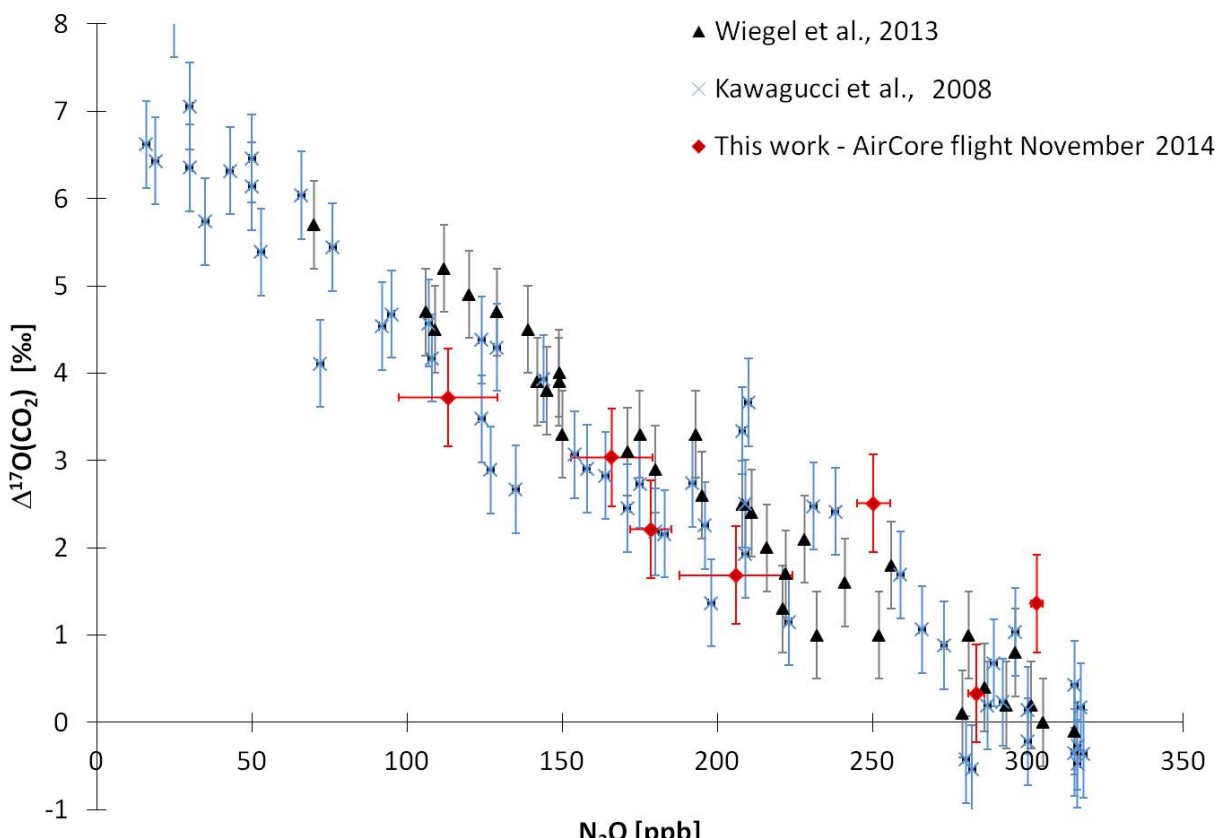

**Figure 8.** Comparison of the linear negative correlation between $\Delta^{17}O(CO_2)$ and $N_2O$ from measurements of the AirCore/SAS samples presented here with literature data from Kawagucci et al. (2008) (crosses) and Wiegel et al. (2013) (triangles). For the AirCore dataset, $N_2O$ mole fractions were deduced from $CH_4$ mole fractions as described in the text. The horizontal error bars are a combination of the range of $CH_4$ mole fractions that is combined within a single SAS segment and the error of the $CH_4$-$N_2O$ transfer function. The vertical error bars represent $\Delta^{17}O(CO_2)$ uncertainty of 0.56 ‰ (1 $\sigma$).





**Table 1.** Statistical improvement of $\Delta^{17}O$ analytical error by combining multiple ($n$) measurements of the same gas into packages and taking the standard error of these packages.

| test group name | number of packages in the group | number of measurements in the package: ($n$) | random error factor: $1/\sqrt{n}$ | SE theoretical: std dev / sqrt($n$) | SE experimental: std over $\Delta^{17}O$ mean/sqrt($n$) |
|---|---|---|---|---|---|
| A | 1 | 270 | 0.061 | 0.074 | |
| B | 2 | 135 | 0.086 | 0.105 | 0.205 |
| C | 3 | 90 | 0.105 | 0.129 | 0.256 |
| D | 5 | 54 | 0.136 | 0.166 | 0.170 |
| E | 6 | 45 | 0.149 | 0.182 | 0.245 |
| F | 9 | 30 | 0.183 | 0.223 | 0.251 |
| G | 10 | 27 | 0.192 | 0.235 | 0.302 |
| H | 15 | 18 | 0.236 | 0288 | 0.316 |
| I | 18 | 15 | 0.258 | 0.315 | 0.345 |
| J | 27 | 10 | 0.316 | 0.386 | 0.405 |
| K | 30 | 9 | 0.333 | 0.407 | 0.472 |
| L | 45 | 6 | 0.408 | 0.498 | 0.531 |
| M | 54 | 5 | 0.447 | 0.546 | **0.570** |
| N | 270 | 1 | 1.000 | 1.220 | **1.220** |