# Peer review of "Stratospheric Air Sub-sampler (SAS) and its application to analysis of $\Delta^{17}O(CO_2)$ from small air samples collected with an AirCore"

_Atmospheric Measurement Techniques, 2016_

## Referee Comment (RC1) · Anonymous Referee #1 · 9 Aug 2016

Mrozek et al. provide new devices for subsampling of the stratospheric air collected by AirCore and analysis of 17O excess of CO2 in the subsamples. The concept of these technology, which directs to easy and low-price observation of stratospheric chemistry, is quite important because rare opportunity to get stratospheric air has prevented us to understand physical transport and chemical reactions occurring there. The focus is suitable for publication on AMT. As I read, authors got successful results from their devices in general. Thus I recommend this manuscript for future publication on AMT, although there are two issues to be solved before the publication as followings.

Major 1: CO2-N2O separation. I consider the GC condition shown in the manuscript (4 mL/min and 40 oC for PoraPLOT Q 25mx0.53mm) is not sufficient for complete

separation between CO2 and N2O. Both the flow rate and oven temperature seem a bit high for suitable setting for the CO2-N2O separation. Actually authors declare only 30 sec separation, and Fig5 seems showing touch between tails of the peaks. I guess it makes worse accuracy and precision, particularly for preCO2. Zoom-up chromatogram of m/z 44-46 on preCO2 analysis is required to claim the complete separation. Or, reasons why authors chose such (strange, as I feel) GC condition should be described. Improvement to get further separation will bring more precise analysis and increase the value of this study.

Major 2: Contamination. Authors regarded serious contamination in 2 of 10 SAS sub-samples and guess a procedure connecting SAS-IRMS system in laboratory as the source of contamination (P11L03-06). In addition, an other SAS subsample was lost due to "accidental instability in He flow." Because this is the manuscript for success-ful development of the SAS-IRMS system, it is serious defect and should be solved. Authors should improve the procedure to avoid the contamination and accident. And results of the test and the revised procedure will be required in the revised version of manuscript.

Minor points:

P03L02-28: These explanation really confused me. A schematic illustration for de-scribing all the instruments and procedure, like flowchart, seems helpful to understand the whole system from AirCore sampling on the site to IRMS analysis in laboratory.

P04L14-16: I feel it is strange. As I read, SAS can get 10 subsamples by each 2m-long tubing. For easy understanding, the description "a 20m-long tubing divided by eleven valves" can be rephrased to "ten 2m-long tubings connected by eleven valves" or like it.

P06L13-15: Can be deleted.

P09L17-25: Section 3.2 is important for this study because authors first use CuO/Ni

system for oxygen exchanging. Results of the test should be shown in table or figure.

P10L07-07: Fig 6?

P10L12-14: Comparison of analytical errors with previous methods actually used for stratospheric air analysis is helpful for readers.

---

## Referee Comment (RC2) · Anonymous Referee #2 · 21 Aug 2016

The paper presents a new Stratospheric Air Sampler which can conserve air retrieved using the AirCore technique, which has already been analysed for CO2 and CH4. This provides a new access route for measurements with analytical instruments that are slow and/or not field deployable. It is a significant advance as other stratospheric air collection techniques are substantially more expensive. The data from the lab experiments and the example flight is of high quality and analysis appears to have been carried out very thoroughly. Some specific and mostly minor comments can be found below.

page 1, line 12: should be "from the SAS"

page 5, line 4 to 5: Please provide quantitative evidence for the reduced peak broad-

ening.

page 5, line 21 to 22: Please provide evidence that the isotopic composition of the CO2 inside the sample loop is not altered due to this exposure to outside air.

page 5, line 26: Please explain the differences to Mrozek et al. (2015). This would also help other sections such as 2.4.2 and 3.1.

page 6, line 12: Please state the purity of the O2. Also, how was the weekly interval determined and is there any evidence that the conversion is quantitative after reconditioning?

page 6, line 18 to 20: Should be "smooths". Also, please explain the mechanism of this smoothing and the composition of the additional 1 ml volume. How much was the precision improved?

page 7, line 30: Should this be 100 s? It doesn't fit with the 80 and 90 s below.

page 8, line 25: Should this be "decrease"?

page 9, line 14: The test confirms the results of Kawagucci et al. (2005). Does it provide any additional evidence?

page 10, line 2: Is this an improvement compared to Mrozek et al. (2015)?

page 10, line 17 and 21: Please give the geographical coordinates.

page 10, line 31: If the balloon burst at 345 km, why does the SAS only contain air from 24.5 km and below?

page 11, line 20: Please provide details of the methane measurements including precisions and how these were translated into the x-axis uncertainties displayed in Figure 8. Also, how can a CH4-N2O correlation from balloon flights in 2009 and 2011 be applied to stratospheric measurements from 2014?

page 12, line 20: This should be CuO/Ni or similar. Please be more consistent throughout the manuscript.

page 12, line 21 to 22: Please consider rewording.

page 12, line 28: Should be "the SAS".

—————————————————————

---

## Author Comment (AC1) · 28 Sep 2016

**Final Response to Reviewer #1**
**MS No.: amt-2016-124**

We thank the Reviewer#1 for helpful and constructive comments.
Our responses (in blue) follow each comment given by Reviewer#1.

(R1#M1) Major point 1:
CO2-N2O separation. I consider the GC condition shown in the manuscript (4 mL/min and 40C for PoraPLOT Q 25mx0.53mm) is not sufficient for complete separation between CO2 and N2O. Both the flow rate and oven temperature seem a bit high for suitable setting for the CO2-N2O separation. Actually authors declare only 30 sec separation, and Fig5 seems showing touch between tails of the peaks. I guess it makes worse accuracy and precision, particularly for preCO2. Zoom-up chromatogram of m/z 44-46 on preCO2 analysis is required to claim the complete separation. Or, reasons why authors chose such (strange, as I feel) GC condition should be described. Improvement to get further separation will bring more precise analysis and increase the value of this study.

We acknowledge that lower temperatures can result in better separation of $N_2O$ from $CO_2$. However, the separation that is achieved at 40 °C is clearly sufficient to eliminate the interference to levels that are well beyond our measurement precision. This was shown in Mrozek et al., (2015), where we also provide more details, including a zoom into the chromatogram. We do not think that it is necessary to repeat this in the present manuscript.

(R1#M2) Major point 2:
Contamination. Authors regarded serious contamination in 2 of 10 SAS subsamples and guess a procedure connecting SAS-IRMS system in laboratory as the source of contamination (P11 lines 3-6). In addition, an other SAS subsample was lost due to "accidental instability in He flow". Because this is the manuscript for successful development of the SAS-IRMS system, it is serious defect and should be solved. Authors should improve the procedure ro avoid the contamination and accident. And results of the test are the revised procedure wll be required in the revised version of manuscript.

The "accidental instability in He flow" that caused loss of one sample was caused by an interruption of the helium supply in the laboratory, which unfortunately happened during the analysis of this sample, but this has nothing to do with the stability of our system. We demonstrate that 7 of the remaining 9 samples were analyzed successfully, and two showed contamination. We think that this may be related to connecting the samples but more statistics may be needed to resolve this issue completely. Nevertheless, we think that our manuscript still proves the successful setup of a system for analysis of $\Delta^{17}O$ in air samples that are provided by the SAS.

(R1#1) P03 lines2-28: These explanation really confused me. A schematic illustration for describing all the instruments and procedure, like a flowchart, seems helpful to understand the whole system from AirCore sampling on the site to IRMS analysis in laboratory.

We include a schematic illustration for describing the overall procedure below:

Figure 1. The overall procedure from AirCore sampling on the site to IRMS analysis in laboratory.

(R1#2) page 4, lines14-16: : I feel it is strange. As I read, SAS can get 10 subsamples by each 2m-long tubing. For easy understanding, the description "a 20m-long tubing divided by eleven valves" can be rephrased to "ten 2m-long tubings connected by eleven valves" or like it.

The relevant section was changed to:
The SAS used for the $CO_2$ isotope measurements in Utrecht is made of ten 2 m long pieces of 1/4 inch diameter stainless steel tubing. The tubings are connected and closed off at the ends by 11 Swagelok valves (part number SS-3CXS4), and bent to form rings. In the following, we refer to the valves as "three-way valves", and the rings as "SAS segments".

(R1#3) page 6, lines13-15: Lines "Under these conditions, the copper metal forms a coating of copper (II) oxide on the surface of the Cu wires according to: $Cu + \frac{1}{2} O_2 \rightarrow CuO$" can be deleated

This sentence describes the reaction that creates CuO coating on Cu/Ni wires. We think that this is relevant, since we are describing the procedure to recondition the surface in this section.

(R1#4) page 9, lines17-25: Section 3.2 is important for this study because authors first use CuO/Ni system for oxygen exchanging results of the test should be shown in table or figure.

We added a figure to show that isotope exchange is complete. The relevant section was changed to:

*"To quantify the efficiency of oxygen isotope equilibration in the CuO oven we* analyzed two samples containing $CO_2$ with very different isotopic composition. The first one was our RefAir, the second one was a synthetic mixture of $RefCO_2$ diluted to 400 ppm $CO_2$ in synthetic air. The samples were injected via a stainless tube that is similar to a segment of the SAS, but longer (4 m length 1/4 inch o.d.). First, the reference air was injected multiple times through this tube. Next, we filled the injection tube with the $RefCO_2$ dilution and continued the measurements. Figure 2 presents the results. The isotopic difference between the two $CO_2$ samples was about 36 ‰ before isotope exchange. After the isotopic exchange reaction both gases were equilibrated to $\delta^{18}O$ = (19.03 ± 0.18) ‰. *From the difference in the oxygen isotopic composition between RefAir and $RefCO_2$ dilution before and after oxygen isotope exchange, the oxygen exchange efficiency in CuO oven was calculated to be >99.5 %. We conclude that the oxygen exchange reaction with CuO/Ni wires at 900 °C is complete."*

[Figure]

Figure 2. Efficiency of oxygen isotope equilibration in the CuO oven. The $\delta^{18}O(CO_2)$ values of RefAir and a mixture of $RefCO_2$ in synthetic air are shown before and after isotope equilibration. The last point of the RefAir measurement sequence after exchange is missing because the peak of the last run was accidentally not registered.

(R1#5) page 10, line 7:  Fig 6?

Yes , correct , "Fig. 7" was changed to "Fig. 6".

(R1#6) page10, lines12-14: Comparison of analytical errors with previous methods actually used for stratospheric air analysis is helpful for readers.

*Page 10 lines 12-14 are extended as follows*:

*"We conclude that repeated measurements on one air sample (up to 54 repetition* that corresponds to 2.6 μmol $CO_2$*) can reduce the uncertainty in $\Delta^{17}O(CO_2)$ to 0.2 ‰. More than 54 repetitions on one air sample improves the $\Delta^{17}O(CO_2)$ uncertainty only marginally.* This uncertainty is in the same range as previously reported techniques for large samples, most of which did not allow the measurement of very small samples. Bhattacharya and Thiemens (1989), reported an uncertainty of 0.1 ‰ using a $BrF_5$ -based technique, Brenninkmeijer and Röckmann (1998) obtained 0.2 ‰ with a two-step fluorination method, Assonov and Brenninkmeijer (2001) reported 0.33 ‰ with a $CeO_2$ exchange method and Mahata et al., 2012 improved this method to an uncertainty of 0.12 ‰. The Kawagucci et al. (2005) method is the only technique that also targeted very small sample sizes (like our system) and they reported an uncertainty of 0.35 ‰.

In our application to the SAS, *for 5 repeated measurements on one stratospheric air sample stored in the SAS* we *expect an uncertainty of 0.57 ‰ for $\Delta^{17}O(CO_2)$ and 0.03 ‰ for both $\delta^{18}O(CO_2)$ and $\delta^{13}C(CO_2)$. This will be compared to the reproducibility of the actual SAS measurements in Sect. 4.2. "*

---

## Author Comment (AC2) · 28 Sep 2016

**Final Response to Reviewer #2**
**MS No.: amt-2016-124**

We thank the Reviewer#2 for helpful and constructive comments.
Our responses (in blue) follow each comment given by Reviewer#2

(R2#1) page 1, line 12: should be "from the SAS"
We agree. The sentence is changed to:
*"For this purpose the analytical system described by Mrozek et al. (2015) was adopted for analysis of air directly from the SAS."*

(R2#2) page 5, line 4 - 5: Please provide quantitative evidence for the reduced peak broadening
The $CeO_2$ grains and the quartz wool to secure the grains from escaping the $CeO_2$ oven lead to a high flow resistance of the isotope exchange unit presented in Mrozek et al. 2015. As a consequence, the equilibrated $CO_2$ peak is very wide and has low intensity (elution period 450 s, maximum intensity of 25 mV) and had to be focused before introduction to the IRMS.

The CuO wires in the new exchange unit exhibit a negligible flow resistance and no quartz wool is needed to hold the wires in place. The equilibrated $CO_2$ peak is therefore directly introduced to the IRMS and takes 60 s to elute.

 The relevant section was changed to:
*"In addition to the requirement of overpressure in the injection unit, another disadvantage of the method presented in  Mrozek et al. (2015) was the severe peak broadening that was introduced by the strong flow resistance of the $CeO_2$ powder in the oxygen exchange unit, which required re-focusing of the $CO_2$ after equilibration. In the new system the powdered $CeO_2$ in the oxygen exchange unit was replaced with CuO wires. As a result the peak broadening was reduced* by a factor of 7.5 (from 450 s to 60 s) *and the equilibrated $CO_2$ could be analysed without re-focusing. In addition, we developed a custom made sample injection unit for samples provided by the Stratospheric Air Sub-sampler (SAS), where both sample and reference gas are injected via the SAS."*

(R2#3) page 5, line 21 - 22: Please provide evidence that the isotopic composition of the CO2 inside the sample loop is not altered due to this exposure to outside air.

We use a flow rate of 1 ml/ min during sample loop loading and completely flush the loop with the sample gas (for 80 s). Since the sample loop has a volume of 1 ml, we flush for 20 s more than what is required to fill the loop with the sample air.  We could not detect any evidence for an influence of outside air. The vent tube attached to Valco valve 1 prevents outside air to affect the air in the sample loop. The vent is made of 30 cm long fused silica capillary.

The relevant section was changed to:
*"A single $\Delta^{17}O(CO_2)$ measurement requires two independent injections of the sample gas. The first injection is used for direct isotope measurement of $CO_2$ (PreCO$_2$), the second injection is for measuring the isotopically equilibrated $CO_2$(PostCO$_2$). We inject the gas (reference air or sample air) into the GC*

*column via a 1 mL sample loop in V1. The sample loop is always filled to ambient pressure because it is open to the outside air via the vent.* We use extended flushing during sample loop loading (the 1 ml sample loop is flushed with the sample gas at a flow rate of 1 ml/min for 80 s ) to avoid interference from outside air that may diffuse through the vent tubing into the sample loop. "

(R2#4) page 5, line 26: Please explain the difference to Mrozek et al. 2015. This would also help other sections such as 2.4.2 and 3.1

We use the same GC as in Mrozek et al. (2015). The manuscript is clarified:

*"This sub-unit consist of a gas chromatography (GC) capillary column (ParaPLOT Q 25 m x 0.53 mm, Varian),* the same as in Mrozek et al. (2015)."

(R2#5) page 6, line 12: Please state the purity of O2. Also, how was the weekly interval determined and is there any evidence that the conversion is quantitative after reconditioning?

We use Ultra High Purity (UHP) quality $O_2$ from Air Products, 99.9992% purity. To sustain the coating of copper (II) oxide on a surface of the Cu wires, we follow the procedure of Kawagucci et al. (2005), who precondition the CuO unit with oxygen on a weekly bases. After CuO oven oxygenation, the IRMS needs at least 50 measurements for the signal to become stable. For this stabilization phase we use air from the RefAir cylinder. During the routine measurements we monitor the $\delta^{18}O(CO_2)$ RefAir vs. VSMOW signal before and after equilibration (see answers to Reviewer 1, Figure 2) to make sure that the oxygen equilibration reaction is complete and highly efficient.

The relevant section is extended to:

*"Before first use, and then on a weekly basis, the CuO oven is conditioned with $O_2$* (Ultra High Purity (UHP), Air Products) *at a flow rate of 20 mL min$^{-1}$ and a temperature of 600 ºC,* following the procedure of Kawagucci et al. (2005). *Under these conditions, the copper metal forms a coating of copper (II) oxide on the surface of the Cu wires according to:*

*Cu+ ½ $O_2$ → CuO.*

After oxygenation, the IRMS needs at least 50 measurements for the signal to stabilize. During routine measurements we monitor the $\delta^{18}O(CO_2)$ RefAir vs. VSMOW signal before and after equilibration to make sure that the equilibration reaction is quantitative."

(R2#6) page 6, line 18 – 20: Should be "smooths". Also, please explain the mechanism of this smoothing and the composition of the additional 1 mL volume. How much was the precision improved?

The additional volume is a stainless steel straight fitting union (Swagelok). The equilibrated $CO_2$ aliquot expands in the inner volume of the Swagelok union and flushes out of this volume smoothly which leads to a smooth peak shape. Regarding the improvement of precision, unfortunately this was not systematically tested, but the reasoning appeared during the development phase of the system,

where several parameters were changed. So it is hard to provide a quantitative number and we have left out this part of the sentence in the revised version.

The sentence is changed to:
 "The loop in *valve V3* is used to insert  *additional volume of*  approximately *1 mL volume* (1/4 inch o.d. tube connected with Swagelok fittings) *into the flow path, which* smooths *and* ensures a compact shape of *the PostCO$_2$ peak* in the IRMS*."*

(R2#7) page 7, line 30: should this be 100 s? It doesn't fit with the 80 and 90 s below.

The timing is correct. It takes 10 s for the AirCore air to travel from the SAS segment to the sample loop and we then flush the loop for 80 s, as described above in R2#3, so the total time is 90 s. At 90 s the loop is fully flushed and full of sample air and Valco valve V1 switches to position INJECT to transfer the first aliquot of the sample into the analytical system. For more clarity the relevant section was changed accordingly:

*It takes 10 s for the AirCore air to* travel from the SAS segment to *the sample loop* and another 80 s to fill it. At *80 s, the flow rate at the MFC is stopped to save sample and the air is allowed to further expand into and fill the injection loop. At 90 s the loop is properly filled with sample air and Valco valve V1 is switched for 40 s to position INJECT to transfer the first aliquot of the sample air into the analytical system.*

(R2#8) page8, line 25: Should this be "decrease"?
We agree. The sentence is changed to:

"Theoretically, we should be able to analyze 9 SA aliquots from each SAS segment (25 mL / 2.7 mL), but as described above the reference air is used as carrier gas and mixes with the sample air. Since we know the isotopic composition of the reference air, the method can potentially be improved by extracting SA information also from the "mixed" SA/RefAir peaks, which may  decrease measurement uncertainty statistically (see Table 1)."

(R2#9) page 9, line 14: The test confirms the results of Kawagucci et al. (2005).  Does it provide any additional evidence?
The sentence is extended to:

*"The test shows that N2O can be effectively removed on CuO/Ni wires at 900 ºC* and confirms the results of Kawagucci et al. (2005). *This removal method can potentially be applied to other trace gas measurement techniques."*

(R2#10) page 10, line 2:  Is this an improvement compared to Mrozek et al. (2015)?

Yes, in the long term stability test  of  Mrozek et al. (2015) method  the standard deviation  over 270 measurement  for $\delta^{18}O(CO_2)$ is 0.16 ‰  and 0.09 ‰ for $\delta^{13}C(CO_2)$.

The relevant section is extended to:
*"In order to investigate the long term stability of our CF-IRMS system and the measurement uncertainty, 540 aliquots of reference air were injected continuously to the CF-IRMS system for 50 hours, comprising 270 individual measurements of the complete isotopic composition of $CO_2$. The standard deviation of $\Delta^{17}O(CO_2)$ over all 270 measurements was 1.22 ‰. This is the error that we assign to a single measurement with the new analytical system. In our previously published method (Mrozek et al., 2015), the $\Delta^{17}O(CO_2)$ standard deviation in such a long term stability test was 1.68 ‰. The improvement compared to the system described in* Mrozek et al. (2015) *is likely due to the replacement of the isotope exchange medium from powdered $CeO_2$ to CuO wires, and through abandonment of a liquid nitrogen trap for re-focusing of the isotopically equilibrated $CO_2$. For $\delta^{18}O(CO_2)$ and $\delta^{13}C(CO_2)$ the standard deviation over all 270 measurements was 0.06 ‰ and 0.07 ‰ respectively versus 0.16 ‰  and 0.09 ‰ in* Mrozek et al. (2015)*. "*

(R2#11) page 10, line 17 – 21: Please give the geographical coordinates.
The descent and sampling started at 67.3472 °N, 26.9317 °E, the landing was at 67.2376 °N, 27.8408 °E.

The relevant section was changed to:
*"The AirCore payload from the University of Groningen was launched about 100 km upstream from Sodankylä, using a meteorological balloon* (67.35 °N, 26.93 °E). *The AirCore was released from the balloon at 34.5 km altitude (at 4.2 hPa) and the coil filled with air during the balloon descent. Time of launch was 10:04 UTC, while the payload landed at 12:40 UTC, thus the total flight duration was 2 hours and 36 minutes. During the balloon ascent the payload traveled south-east towards Sodankylä, due to winds in the stratosphere and troposphere. The payload landed on a parachute 53.8 km east from the analysis laboratory of the Finnish Meteorological Institute (FMI),* 67.24 °N, 27.84 °E. *"*

(R2#12) page 10, line 31: If the balloon burst at 34.5 km, why does the SAS only contain air from 24.5 km and below?
Before each flight the AirCore coil is filled with fill air, which mixes the highest part of the stratospheric air samples. To make sure no fill-gas was sampled into the SAS, we intentionally discarded a fraction of the stratospheric air samples. For this particular flight on Nov 5 2014, we retrieved AirCore profiles up to 30 km (7 mbar) after removing the contaminated samples. Unfortunately, the stratospheric air samples collected between 30 km and 25 km (7 - 20 mbar) were flushed away, so that the SAS segment 1 effectively sampled the stratospheric air samples that were collected between 24.5 and 21.4 km (22 - 38 mbar). We have recently improved the timing of sub-sampling to discard less stratospheric air samples than previously.

Section 4.1 is extended:

*We assigned the trace gas mixing ratios measured with the Picarro instrument to the individual SAS segments based on the flow rate of the carrier gas and time required to fill the SAS. Figure 7 shows which part of the stratospheric AirCore air is stored in which segment of the SAS, both as function of altitude and pressure. Note, that the SAS segments are equally spaced in pressure change (here 16 mbar/segment) during the descent rather than altitude. For example: the SAS segment 1 contains air from 24.5 to 21.4 km, while the SAS segment 2 contains air from 21.4 to 19.2 km, etc.* SAS does not contain the air from the balloon highest altitude (~34.5 km) because the highest part of the stratospheric air samples was not collected during the sub-sampling into SAS procedure. To make sure no fill-gas was sampled into the SAS, we intentionally discarded a fraction of the stratospheric air samples. For this particular flight on Nov 5 2014, we retrieved AirCore profiles up to 30 km (7 mbar) after removing the contaminated samples. Unfortunately, the stratospheric air samples collected between 30 km and 25 km (7 - 20 mbar) were flushed away, so that the SAS segment 1 effectively sampled the stratospheric air samples that were collected between 24.5 and 21.4 km (22 - 38 mbar). We have recently improved the timing of sub-sampling to discard less stratospheric air samples than previously. *An analysis of the trace gas profiles from the AirCore flight will be published separately.*

(R2#13) page 11, line 20: Please provide details of the methane measurements including precisions and how these were translated into the x axis uncertainties displayed in Figure 8. Also how can a CH4-N2O correlation from balloon flights in 2009 and 2011 be applied to stratospheric measurements from 2014?

The horizontal errors in Figure 8 are based on the standard deviation over the Picarro measurements of $CH_4$ mole fraction for each individual SAS segment. These are converted to errors in $N_2O$ using the fit function from the cryosampler flight (Engel et al, 2015). We choose the fit function from the cryosampler because this is the dataset that is closest in time and space to our samples that was available to us. The cryosampler reached an altitude of 34 km and therefore the correlation covers stratospheric altitudes of our interest. We are presently preparing a manuscript on the scientific interpretation of the AirCore data together with aircraft data, which will deal with the conversion from $CH_4$ in $N_2O$ in much more detail and prefer to keep the present technical manuscript focused on the isotope technique.

The first paragraph of section 4.3 is extended:

*The mole fraction of $N_2O$ is a good tracer for the photochemical processing of long-lived trace gases in the stratosphere (Park et al., 2004, Kaiser et al., 2006). $N_2O$ was not measured on the AirCore air, but $CH_4$ was measured and we use the relationship between stratospheric $CH_4$ and $N_2O$ mole fractions at high latitudes to translate the $CH_4$ mole fractions to $N_2O$ mole fractions. For this purpose we used $CH_4$ – $N_2O$ data from two cryosampler flights in the Arctic in 2009 and 2011 (Engel et al., 2015). The fit function for the $CH_4$ – $N_2O$ data set was provided to us by Andreas Engel, Goethe University Frankfurt, Germany. The stratospheric pseudo-$N_2O$ profile derived this way was then averaged according to the ten individual SAS segments.* A detailed discussion on obtaining $N_2O$ mole fractions from the measured $CH_4$ will be provided together with the scientific interpretation of the data in (Mrozek et al., Triple oxygen isotope variations as a tracer of polar vortex air, manuscript In preparation).

(R2#14) page 12, line 20: This should be CuO/Ni or similar. Please be more consistent throughout the manuscript.

We agree. The sentence is changed to:

"*The determination of Δ$^{17}$O(CO$_2$) is performed by measuring CO$_2$ before and after complete oxygen isotope exchange with a large oxygen reservoir provided by CuO/Ni .*"

(R2#15) page 12, line 21 -22: Please consider rewording in the sentence "Since no focusing of isotopically exchange CO$_2$ is needed the analytical system operates liquid nitrogen free."

The sentence is changed to:

The analytical system operates liquid nitrogen free because the isotopically equilibrated CO$_2$ does not require a focusing step, unlike the Mrozek et al. 2015 method.

(R2#16) page 12, line 28: Should be "the SAS".

We agree. The sentence is changed to:

"*The concept of the SAS will further broaden the scientific questions that can be addressed by AirCore sounding (e.g. Paul et al. (2016)).*"